# Twenty-Four-Hour Compositional Data Analysis in Healthcare: Clinical Potential and Future Directions

**DOI:** 10.3390/ijerph22071002

**Published:** 2025-06-25

**Authors:** Cain Craig Truman Clark, Clarice Maria de Lucena Martins

**Affiliations:** 1College of Life Sciences, Birmingham City University, Birmingham B15 3TN, UK; 2Laboratory for Integrative and Translational Research in Population Health, Research Centre in Physical Activity, Health and Leisure, University of Porto, 4200-450 Porto, Portugal; clarice@fade.up.pt

**Keywords:** Compositional Data Analysis, epidemiology, behavior, time use

## Abstract

Compositional Data Analysis (CoDA) is a powerful statistical approach for analyzing 24 h time-use data, effectively addressing the interdependence of sleep, sedentary behavior, and physical activity. Unlike traditional methods that struggle with perfect multicollinearity, CoDA handles time use as proportions of a whole, providing biologically meaningful insights into how daily activity patterns relate to health. Applications in epidemiology have linked variations in time allocation between behaviors to key health outcomes, including adiposity, cardiometabolic health, cognitive function, fitness, quality of life, glycomics, clinical psychometrics, and mental well-being. Research consistently shows that reallocating time from sedentary behavior to sleep or moderate-to-vigorous physical activity (MVPA) improves health outcomes. Importantly, CoDA reveals that optimal activity patterns vary across populations, supporting the need for personalized, context-specific recommendations rather than one-size-fits-all guidelines. By overcoming challenges in implementation and interpretation, CoDA has the potential to transform healthcare analytics and deepen our understanding of lifestyle behaviors’ impact on health.

## 1. Introduction

The understanding of human health and well-being has increasingly recognized the critical role of daily activity patterns, encompassing the interplay between sleep, sedentary behavior, and physical activity [1,2]. These components of a 24 h cycle are not independent entities but are intrinsically linked, where time allocated to one behavior directly influences the time available for others [3]. For instance, dedicating more time to physical activity necessitates a reduction in time spent sleeping or being sedentary. This interconnectedness suggests that analyzing these behaviors in isolation may provide an incomplete or misleading picture of their impact on health outcomes [4]. Traditional statistical methods, often designed for analyzing independent variables, struggle to effectively capture the complexities arising from the co-dependent nature of time-use data [3]. The perfect multicollinearity inherent in 24 h time budgets further complicates the application of conventional multivariate analyses [5].

To address these limitations, Compositional Data Analysis (CoDA) has been utilized as a robust statistical framework specifically designed for analyzing data that represent parts of a whole, such as the 24 h time budget [6]. CoDA acknowledges the relative nature of time-use data, focusing on the proportions or ratios between different activities rather than their absolute durations. This approach offers a statistically sound and biologically relevant method for investigating the intricate relationships between daily activity patterns and various health outcomes [3]. The increasing adoption of CoDA in time-use epidemiology reflects a growing recognition within the research community of the need for analytical techniques that respect the compositional nature of human behavior across the day [7,8,9]. By providing a better understanding of how individuals allocate their time, 24 h CoDA holds significant clinical potential for informing personalized interventions, developing targeted health recommendations, and ultimately improving patient management and prevention strategies.

The recognition that time spent in different activities within a 24 h period is not independent is fundamental to understanding human physiology and behavior. Because time is a finite resource, any increase in one activity invariably leads to a decrease in another. Analyzing these shifts and their subsequent health consequences necessitates methods capable of accounting for this interdependence, a capability often lacking in traditional analytical approaches. The growing prevalence of CoDA in time-use epidemiology indicates a significant shift towards more statistically rigorous and biologically meaningful analyses. Researchers are increasingly employing methods that are specifically tailored to handle the compositional nature of their data.

## 2. The Principles of Compositional Data Analysis in Time-Use Epidemiology

The foundation of CoDA rests on several key statistical principles. Compositional data are defined as multivariate data where the components represent parts of a whole and sum to a constant value, such as the 24 h in a day [3]. In time-use epidemiology, these components typically include sleep, sedentary behavior, and physical activity of varying intensities [10,11], with this being exemplified in the Canadian 24 h Movement Guidelines and accompanying infographics (see here: https://csepguidelines.ca/ (accessed on 18 June 2025)). A central tenet of CoDA is the focus on relative information [6]. The absolute amount of time spent in a particular activity is less meaningful in isolation than its proportion relative to the total time and other activities. For instance, spending 30 min in moderate-to-vigorous physical activity carries different implications if the rest of the day is spent sleeping versus engaging in prolonged sedentary behavior.

Compositional data reside in a constrained geometric space known as the Simplex [12]; simply, they exist within a limited space where all parts must add up to a whole. For example, in a three-part composition (such as sleep, sedentary behavior, and physical activity), this can be visualized as a triangle, or ternary plot, where each point within the triangle represents a unique combination of the three components that sum to the total (example three-part composition figure is depicted in [3]). Standard statistical methods, however, assume data can exist anywhere in real space, making them ill-suited for the analysis of compositional data within the Simplex space [12]. To overcome this, CoDA employs log-ratio transformations to transform the compositional data from the constrained Simplex space into an unconstrained Euclidean space, allowing for the application of traditional multivariate statistical methods [12]. Common log-ratio transformations include the additive log-ratio (alr), centered log-ratio (clr), and isometric log-ratio (ilr) [3]. Isometric log-ratio transformations, in particular, preserve the geometric relationships between the data points, making them highly suitable for multivariate analyses like regression [13].

The advantages of using CoDA over traditional methods in time-use epidemiology are numerous [3]. CoDA allows for the simultaneous inclusion of all components of the 24 h time budget in statistical models without encountering issues of multicollinearity [3]. By focusing on the relative allocation of time, CoDA provides a more accurate and biologically relevant understanding of how different patterns of daily activities are associated with health outcomes [11,14,15]. Furthermore, CoDA enables the quantification of the joint effects of all movement behaviors and allows for the estimation of changes in health outcomes associated with reallocating time between different activities [16]. This capability is particularly valuable for informing targeted interventions and developing integrated health recommendations [5].

## 3. Applications of 24-h CoDA in Understanding Health Outcomes

The application of 24 h CoDA has yielded significant insights into the relationship between daily activity patterns and a wide range of physical health outcomes [17,18]. In the realm of adiposity and weight management, numerous studies have employed CoDA to demonstrate the importance of the balance between sleep, sedentary behavior, and physical activity [19,20,21]. It has been evidenced that spending more time sleeping and engaging in moderate-to-vigorous physical activity (MVPA) relative to other behaviors is associated with lower adiposity [22,23]. Research on Malaysian children found that replacing sedentary behavior and light physical activity with either sleep or MVPA was linked to reduced adiposity [24]. Interestingly, increased sleep has been shown to be associated with reduced BMI, highlighting the potentially complex interplay between these behaviors [25]. Furthermore, the pattern of physical activity accumulation appears to be important, with evidence suggesting that accumulating light-intensity physical activity in short bursts may be more beneficial for limiting adiposity in children [26].

CoDA has also been instrumental in understanding the links between 24 h activity compositions and cardiometabolic health. A study in Australian participants showed that the overall time-use composition was associated with markers of cardiometabolic health [27]. Research on children and youth in Norway found that vigorous physical activity was negatively associated with BMI and waist circumference, while sedentary time showed a positive association with waist circumference [28]. The “Goldilocks Day” concept, explored through CoDA, has provided valuable insights into the optimal durations of sleep, sedentary time, light physical activity, and MVPA for children’s skeletal health, generally indicating that more sleep and MVPA, and less sedentary time, are beneficial for bone health [29]. In the context of mortality, studies using CoDA have demonstrated a significant association between the proportion of time spent in MVPA and a lower risk of all-cause mortality [30,31,32]. The overall composition of daily activities, including sleep, sedentary behavior, and physical activity, has also been shown to be associated with mortality rates [33]. Moreover, research on preschoolers has indicated that the 24 h movement behavior composition is a more important predictor of fundamental movement skills than any single type of movement considered in isolation [34,35].

The application of CoDA has extended to the investigation of mental health and cognitive outcomes. Studies have explored the relationship between 24 h activity compositions and indicators such as psychological distress, work engagement, and emotional and behavioral problems. For instance, research on left-behind children in China found that reallocating time from sedentary behavior to sleep and light physical activity was associated with improvements in internalizing and externalizing problems [36]. In overweight and obese college students, negative associations were observed between depression, anxiety, and stress with time spent in MVPA and sleep, while positive associations were found with screen-based and non-screen-based sedentary behavior [37]. Among Japanese workers, time spent sleeping was positively correlated with mental health indicators, whereas time spent in sedentary behavior or light physical activity showed negative correlations [38]. Regarding cognitive function and academic performance, studies using CoDA in Asian children revealed that while the overall 24 h movement behavior composition might not be directly associated with executive function and academic achievement, specific reallocations of time, such as from MVPA to sleep, could lead to improved academic performance [39].

CoDA has also been applied to understand health outcomes in specific populations and contexts. This includes research on children across various age groups, older adults, workers, overweight and obese individuals, and retirees. Notably, studies have examined the influence of socioeconomic status on children’s time-use compositions, revealing significant differences in how children from different socioeconomic backgrounds allocate their time [40]. For example, children from higher-socioeconomic-status families tend to spend more time on school-related activities and less time on screen use and sleep compared to those from lower-socioeconomic-status families [40]. Furthermore, cultural and regional differences in 24 h movement behaviors and their associations with health have been explored across various countries, including China, Japan, South Korea, Australia, New Zealand, Singapore, and South Africa [41,42,43,44,45,46,47]. These studies highlight the importance of considering the Human Development Index (HDI) of different regions when developing global guidelines and acknowledge the specific challenges of implementing such guidelines in low- and middle-income countries. Cultural factors have also been shown to significantly influence children’s sleep patterns [48].

We perceive several clear themes from the application of CoDA. First, health outcomes are influenced not by individual behaviors alone, but by the relative balance between sleep, sedentary time, and physical activity. Second, reallocating time from sedentary behavior to either sleep or moderate-to-vigorous physical activity (MVPA) is consistently linked to better physical, mental, and cognitive health. Third, CoDA reveals that patterns may differ across populations; indeed, factors such as age, socioeconomic status, and culture shape how time is used and its impact on health. Finally, the evidence suggests that personalized and context-specific recommendations are likely to be more effective than universal, one-size-fits-all guidelines.

## 4. Clinical Potential and Implications of CoDA

The findings derived from 24 h CoDA hold significant clinical potential for transforming healthcare practices. One key implication lies in informing personalized interventions. By analyzing an individual’s unique time-use composition and understanding its association with their specific health goals, clinicians can tailor activity recommendations that go beyond generic advice. For instance, if an individual aims to improve their mental well-being, CoDA findings might suggest reallocating time from sedentary screen use to both sleep and light physical activity, providing a more nuanced and effective strategy than simply recommending more exercise. Numerous studies have demonstrated the potential for time reallocation between different movement behaviors to improve various health outcomes, offering a data-driven approach to personalized lifestyle modifications [49]. For instance, in clinical glycomics, the use of CoDA as a framework has been asserted to reveal insights into glycome variations, critical to understanding roles of glycans in health and disease [50].

CoDA findings are also crucial for developing targeted health recommendations and guidelines. Indeed, the shift towards integrated 24 h movement guidelines, which acknowledge the interconnectedness of sleep, sedentary behavior, and physical activity, is largely supported by the evidence emerging from CoDA research [51]. These guidelines aim to provide a more holistic approach to promoting healthy lifestyles. Furthermore, CoDA can inform the development of age-specific and population-specific guidelines [52]. Research has highlighted the need for guidelines to be culturally relevant and inclusive of underrepresented regions [52]. The observation that the relationship between health-related quality of life and MVPA in children varies based on a country’s HDI underscores the importance of considering broader societal factors when formulating global recommendations [53].

In the milieu of patient management and prevention strategies, CoDA offers valuable tools for identifying unhealthy time-use patterns that may be associated with an increased risk of disease. Indeed, by understanding these patterns, healthcare professionals can develop targeted interventions aimed at promoting healthier daily routines. Moreover, CoDA can be used to monitor the impact of interventions on an individual’s overall 24 h activity composition, providing a comprehensive assessment of behavioral changes [6]. While the provided examples on children did not extensively cover the direct application of CoDA in disease diagnosis and prognosis, the broader methodology has demonstrated potential in these areas within healthcare, particularly in fields like microbiome analysis and biomarker discovery [54]. Of additional potential is the revolution of clinical psychometrics; indeed, Lehmann and Vogt [55] advocate for the universality and reliability of the CoDA approach in psychometric big data analysis affecting psychometric health economics, patient welfare, grant funding, economic decision-making, and profits.

The power of CoDA lies in its ability to move healthcare beyond simplistic, isolated recommendations for individual behaviors towards more personalized and integrated strategies for health promotion and disease prevention. By understanding the distinct time-use compositions associated with various health outcomes in diverse populations, clinicians can craft more effective and tailored interventions. The capacity of CoDA to inform the development of 24 h movement guidelines that recognize the interdependence of behaviors marks a significant advancement in public health recommendations. Traditional guidelines often treat sleep, physical activity, and sedentary behavior as separate entities. CoDA provides the evidence base for more holistic and synergistic approaches to promoting healthy lifestyles.

Additionally, CoDA has potentially significant applications in clinically relevant contexts, outside of time use, particularly in fields where variables are inherently constrained by their relative proportions. In nutritional epidemiology, CoDA has been employed to analyze dietary intake patterns, recognizing that macronutrient consumption is not independent, but rather part of a closed system [56,57]. This approach enables researchers to assess how the relative distribution of carbohydrates, fats, and proteins influences disease risk, such as in metabolic disorders [58]. Similarly, CoDA has been applied in microbiome research, where microbial communities are best understood through their compositional nature rather than absolute abundances [59]. For instance, shifts in the proportional abundance of gut microbiota have been linked to conditions such as inflammatory bowel disease and obesity [60]. Moreover, CoDA has proven useful in biomarker analysis, where the balance of molecular indicators within biological samples can provide insights into disease progression or treatment response [61]. By addressing the inherent interdependencies within compositional datasets, CoDA can be leveraged to improve the validity of statistical analyses and enhance clinical decision-making.

## 5. Challenges, Limitations, and Future Directions

Despite the significant potential of 24 h CoDA in healthcare, several methodological and practical barriers must be addressed before its widespread adoption can occur. Methodologically, handling zero values remains a core challenge, as log-ratio transformations, fundamental to CoDA, are undefined for zeros, requiring imputation or amalgamation approaches that may introduce bias or uncertainty [62]. While innovative solutions such as curvature analysis have been proposed to address this [63], no consensus approach exists. Additionally, the interpretation of log-ratio coefficients is often complex and requires specialized statistical expertise, which can hinder understanding and acceptance by clinicians and public health practitioners [62].

Beyond technical issues, key barriers to implementing CoDA in clinical practice include limited clinician familiarity with compositional data concepts, analyses, and outcomes. Many healthcare professionals lack training in these advanced statistical methods, likely reducing confidence in applying or interpreting CoDA results. Moreover, the current lack of accessible, user-friendly software tools tailored for clinical or epidemiological use impedes routine analysis and decision-making based on compositional data. Data constraints also pose challenges; for example, accelerometer devices or questionnaires used to capture 24 h movement behaviors can have limitations in accuracy and may not capture important contextual information about activity quality or type, further complicating translation to practice [64,65].

While the section on future directions highlights promising integration areas such as machine learning and personalized medicine, elaboration is needed on how these can be realized. Indeed, integrating CoDA with machine learning techniques could enhance pattern recognition and predictive modeling by accommodating the compositional nature of time-use data. However, such integration requires the development of new algorithms capable of handling compositional constraints without violating their geometric properties [66]. Similarly, applying CoDA in personalized medicine requires tailored models that consider individual variability in daily behavior patterns and their health impacts. Translational challenges include validating these models in diverse populations, ensuring interpretability for clinicians, and embedding results into clinical workflows [67].

To overcome these barriers, future research should focus on multidisciplinary collaborations between statisticians, clinicians, and data scientists to develop comprehensive training programs and user-friendly analytical platforms that facilitate CoDA usage. Expanding longitudinal and intervention studies will also strengthen causal inferences and demonstrate clinical utility, fostering clinician trust [51,68]. Additionally, exploring how the context and quality of activities influence health outcomes within the compositional framework will provide richer, actionable insights [68,69,70]. Addressing these methodological and practical challenges is essential to fully realize CoDA’s promise in delivering personalized, effective healthcare interventions.

## 6. Conclusions

CoDA provides a powerful, statistically rigorous framework for analyzing how daily behaviors interact within the 24 h cycle. Its strength lies in capturing the relative balance of sleep, sedentary time, and physical activity, revealing complex links with diverse health outcomes, from adiposity, glycomics, and cardiometabolic markers to mental health, cognition, and clinical psychometrics, across varied populations. Clinically, CoDA offers a promising path toward personalized, data-driven interventions and integrated 24 h guidelines that reflect the synergistic nature of time-use behaviors. However, key challenges, such as handling zeros, interpreting complex log-ratio outputs, and proving causal relationships, must be addressed to unlock its full potential. Continued methodological innovation, better clinician education, and integration with emerging tools, like machine learning and personalized medicine, will be critical to translating CoDA insights into practical healthcare strategies that improve health and well-being throughout the life course.

## Data Availability

No new data were created or analyzed in this study.

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
