# Peer review of "Twenty-Four-Hour Compositional Data Analysis in Healthcare: Clinical Potential and Future Directions"

_ijerph, 2025, doi:10.3390/ijerph22071002_

Round 1
Reviewer 1 Report
Comments and Suggestions for Authors
Thank you for the opportunity to review the manuscript titled: “24-Hour Compositional Data Analysis in Healthcare: Clinical Potential and Future Directions”, Manuscript ID: ijerph-3667079. The manuscript addresses the emerging and valuable application of Compositional Data Analysis (CoDA) in healthcare. The manuscript is timely, well-structured, and demonstrates the authors’ strong familiarity with the statistical framework and its interdisciplinary applications.
However, several weaknesses limit the current impact and practical utility of the paper. I outline the key areas of concern below:
Several sections, particularly in the introduction and principles, are overly verbose and include redundant phrasing (e.g., “inherent interconnectedness” and “multicollinearity”). Simplifying the language and reducing repetition would significantly enhance readability, especially for non-technical readers or those new to CoDA.
While the manuscript thoroughly discusses the theoretical and statistical basis of CoDA, it remains largely abstract in its clinical implications. The discussion would be substantially strengthened by including concrete examples or pilot studies where CoDA has been implemented in clinical or public health practice. This would help contextualize the clinical “potential” repeatedly mentioned throughout the paper.
Please consider including missed opportunities for synthesis.The review of studies in the “Applications” section is rich but somewhat fragmented. The authors cite many examples in succession without drawing thematic or comparative conclusions. A more integrative synthesis (e.g., summary tables or comparative insights across age groups or health domains) would provide greater clarity and narrative flow.
Please consider more discussion of barriers to implementation. Although the manuscript identifies certain technical challenges (e.g., handling zeros, interpreting log-ratio coefficients), it does not sufficiently explore the barriers to CoDA adoption in clinical practice. Issues such as clinician unfamiliarity, software limitations, or data constraints should be discussed in more detail.
The section on future directions lists promising areas (e.g., machine learning, personalized medicine), but lacks depth. Please consider to elaborate on how such integration could be achieved, and what specific methodological or translational challenges remain.
Thank you again for the opportunity to review this work.
Author Response
C1: Thank you for the opportunity to review the manuscript titled: “24-Hour Compositional Data Analysis in Healthcare: Clinical Potential and Future Directions”, Manuscript ID: ijerph-3667079. The manuscript addresses the emerging and valuable application of Compositional Data Analysis (CoDA) in healthcare. The manuscript is timely, well-structured, and demonstrates the authors’ strong familiarity with the statistical framework and its interdisciplinary applications.
R1; We thank the reviewer for their comments on out perspectives manuscript. We have sought to incorporate comments thoughtfully throughout the manuscript.
C2: Several sections, particularly in the introduction and principles, are overly verbose and include redundant phrasing (e.g., “inherent interconnectedness” and “multicollinearity”). Simplifying the language and reducing repetition would significantly enhance readability, especially for non-technical readers or those new to CoDA.
R2: We have carefully proofread the manuscript and reduced the amount of potential redundancy. Some terms, like multicollinearity, are retained due to their specific statistical meaning.
C3: While the manuscript thoroughly discusses the theoretical and statistical basis of CoDA, it remains largely abstract in its clinical implications. The discussion would be substantially strengthened by including concrete examples or pilot studies where CoDA has been implemented in clinical or public health practice. This would help contextualize the clinical “potential” repeatedly mentioned throughout the paper.
R3: We have now provided some additional examples, more specific to clinical implication. For instance, we highlighted applications in clinical glycomics and clinical psychometrics. Both examples are contained in the ‘Clinical Potential and Implications of CoDA’.
C4: Please consider including missed opportunities for synthesis. The review of studies in the “Applications” section is rich but somewhat fragmented. The authors cite many examples in succession without drawing thematic or comparative conclusions. A more integrative synthesis (e.g., summary tables or comparative insights across age groups or health domains) would provide greater clarity and narrative flow.
R4: We appreciate this comment. We have elected to draw together themes, from our perspective (as the paper type is), and ended the paragraph with this.
C5: Please consider more discussion of barriers to implementation. Although the manuscript identifies certain technical challenges (e.g., handling zeros, interpreting log-ratio coefficients), it does not sufficiently explore the barriers to CoDA adoption in clinical practice. Issues such as clinician unfamiliarity, software limitations, or data constraints should be discussed in more detail. The section on future directions lists promising areas (e.g., machine learning, personalized medicine), but lacks depth. Please consider to elaborate on how such integration could be achieved, and what specific methodological or translational challenges remain.
R5: We have combined the 2 comments above, so we have now included additional information surrounding barriers to adoption. We also included some perspectives on addressing certain issues (e.g., zero handling). We do contend however, that as this article type is ‘perspective’, we are not able to provide detailed ‘review’ scale information.
Reviewer 2 Report
Comments and Suggestions for Authors
Dear authors,
This manuscript addresses an increasingly important topic in healthcare analytics by presenting Compositional Data Analysis (CoDA) as a robust framework for understanding 24-hour time-use data. The paper is conceptually strong and provides a timely synthesis of current literature. This work has importance to clinical and public health audiences is evident, and the discussion of potential applications across various domains is also a point of importance.
After read the manuscript, there are several areas where it would benefit from further clarification and development to strengthen its impact:
- Clarify Technical Terminology: Some statistical language—particularly around “Simplex space” and “log-ratio transformations”—is presented in a way that may be inaccessible to readers without a background in CoDA. Consider rephrasing or briefly explaining these terms in more intuitive language. A simple example or analogy could also be helpful.
- Deepen the Literature Synthesis: This manuscript lists multiple studies in several sections with a descriptive aproach. In my point of view it limits the interpretive strength of the narrative. The authors should focus on a few pivotal studies and provide more critical commentary on their design, sample, findings, and relevance. This aproach would make the conclusions more persuasive and grounded.
- Include Practical or Visual Illustrations: As the manuscript makes a strong case for the clinical applicability of CoDA, a short illustrative scenario or schematic diagram showing how CoDA might inform real-world decision-making (e.g., time reallocation in a patient’s daily routine) would enhance reader understanding and bridge theory with practice.
- Expand on Methodological Challenges: The manuscript briefly acknowledges the challenges associated with handling zero values and imputing data in Compositional Data Analysis, but this important topic should be discussed whit more detail.
- Refine Repetitive Language and Improve Transitions: Some key concepts (e.g., “interdependence,” “holistic approach,” and “biologically meaningful”) are repeated across multiple sections. You may wish to revise for conciseness and improve the flow between sections for a smoother reading experience.
- Add Figures or Tables Where Appropriate: Including at least one figure—such as a diagram of the 24-hour movement composition or a hypothetical time-use shift modelled with CoDA—would not only strengthen the presentation but also help readers unfamiliar with the methodology visualize the key concepts.
Once these revisions are addressed, the manuscript will be well-positioned to contribute meaningfully to the field.
Comments on the Quality of English LanguageThe manuscript is generally well-written, but some sentences could benefit from tighter phrasing and improved clarity.
Author Response
C1: This manuscript addresses an increasingly important topic in healthcare analytics by presenting Compositional Data Analysis (CoDA) as a robust framework for understanding 24-hour time-use data. The paper is conceptually strong and provides a timely synthesis of current literature. This work has importance to clinical and public health audiences is evident, and the discussion of potential applications across various domains is also a point of importance.
R1: Thank you for your overall and specific comments. We appreciate your thoughts, and have sought to incorporate your suggestions accordingly.
C2: Clarify Technical Terminology: Some statistical language—particularly around “Simplex space” and “log-ratio transformations”—is presented in a way that may be inaccessible to readers without a background in CoDA. Consider rephrasing or briefly explaining these terms in more intuitive language. A simple example or analogy could also be helpful.
R2: We have sought to provide some additional definition/example to statistical terminology in the manuscript
C3: Deepen the Literature Synthesis: This manuscript lists multiple studies in several sections with a descriptive aproach. In my point of view it limits the interpretive strength of the narrative. The authors should focus on a few pivotal studies and provide more critical commentary on their design, sample, findings, and relevance. This aproach would make the conclusions more persuasive and grounded.
R3: Given the perspective nature of the manuscript, we have retained a broad literature base; however, we have provided better detail in a few pivotal studies (again, from our perspective). We also provided overarching themes, from our perspective, so these can be “taken away” more readily.
C4: Include Practical or Visual Illustrations: As the manuscript makes a strong case for the clinical applicability of CoDA, a short illustrative scenario or schematic diagram showing how CoDA might inform real-world decision-making (e.g., time reallocation in a patient’s daily routine) would enhance reader understanding and bridge theory with practice.
R4: Thank you, this is a very good suggestion. We have incorporated this suggestion into the manuscript.
C5: Expand on Methodological Challenges: The manuscript briefly acknowledges the challenges associated with handling zero values and imputing data in Compositional Data Analysis, but this important topic should be discussed whit more detail.
R5: This section has been expanded accordingly, with some offering of answers provided. Though the caveat remains that it is an on-going challenge in compositional analyses.
C6: Refine Repetitive Language and Improve Transitions: Some key concepts (e.g., “interdependence,” “holistic approach,” and “biologically meaningful”) are repeated across multiple sections. You may wish to revise for conciseness and improve the flow between sections for a smoother reading experience.
R6: Revision for conciseness has been undertaken; thank you for your suggestion
C7: Add Figures or Tables Where Appropriate: Including at least one figure—such as a diagram of the 24-hour movement composition or a hypothetical time-use shift modelled with CoDA—would not only strengthen the presentation but also help readers unfamiliar with the methodology visualize the key concepts.
R7: We have sought to provide updated examples, a compilation of key themes, and reference to existing exemplar infographics, namely the Canadian 24 movement guidelines, which showcases the 24h approach with great clarity. Given the ‘perspective’ nature of the article, we elected not to create a bespoke image.